# Benthic Resource Baseline Mapping of *Cakaunisasi* and *Yarawa* Reef Ecosystem in the Ba Region of Fiji

**Ashneel Ajay Singh** [1,2,*], **Anish Maharaj** [3] **and Priyatma Singh** [2]

1    School of Agriculture, Geography, Environment, Ocean and Natural Sciences,
     The University of the South Pacific, Suva 679, Fiji
2    Department of Science, School of Science and Technology, The University of Fiji, Lautoka 679, Fiji;
     priyatmas@unifiji.ac.fj
3    SpatialWorks, Queens Rd, Lautoka 679, Fiji; maharajanish0323@gmail.com
*    Correspondence: singh_ah@usp.ac.fj; Tel.: +679-323-2581

**Abstract:** Coastal habitats form a critical source of livelihood for a large number of inhabitants in Fiji. The absence of historical and baseline information creates a significant challenge in effectively designing suitable management plans. This study aimed at developing reliable benthic cover maps of village intertidal resource harvest areas (Cakaunisasi and Yarawa reefs) and anthropogenic perceptions of Votua Village in the Ba region of Fiji for better resource management planning and monitoring. Images captured by the WorldView2 satellite were used as a base for mapping out the resources. Data logging on-site, Global Positioning System (GPS) recordings, local interviews and high-resolution video capturing were utilised for ground-truthing techniques. Six classes of benthic cover were identified, which included algae, coral, sand and gravel, buried reef, coral rubble and seagrass. Accuracy assessment and supervised classification were done using ground reference points. There was an existing marine protected area (MPA) on the Yarawa reef, which did not seem to be working as well as anticipated by observing the habitat maps of the two reefs. Baseline maps constructed here and possibly ecosystem maps can allow for monitoring of the existing MPA as well as the formation of a new and more informed MPA. The maps generated in this study serve as baseline information about resource distribution on Cakaunisasi and Yarawa reefs to inform management decisions.

**Keywords:** GIS; Votua; coastal resource; benthic cover; supervised classification; accuracy assessment; WorldView2; habitat map; satellite imagery

## 1. Introduction

Pacific Island countries (PICs) consist of mostly small developing states, with a large proportion of the population having heavy dependence on coastal resources for their subsistence and socioeconomic livelihood. With an increase in the population, combined with anthropogenic effects and the changing climate over time, the pressure on coastal resources poses a threat to their continual sustainability. Most coastal resource users keep very little to no record of their harvest. The absence of historical and baseline information creates knowledge gaps and poses a significant challenge in effectively assessing the state of coastal resources and designing suitable management plans.

This is true for Fiji, where coastal habitats form a critical source of livelihood for a large number of inhabitants. The local perception seems to assume the changing climate as the major or the only factor posing a negative influence on coastal habitats. Such perceptions are dangerous as coastal resources are influenced by various factors, including coastal land use activities, over-exploitation, coastal tourism, deforestation and anthropogenic waste disposal along the inland watersheds leading to the coasts. The interaction and balance of various factors and conditions result in narrow optimum conditions that are critical to the survival and thriving of coastal habitats [1]. Coral reef habitats are critical to coastal

habitats, and their indiscriminate and unplanned use are causing rapid degradation of the resources [2,3]. Research works have shown that the distribution and abundance of coastal fishery resources are considerably influenced by anthropogenic effects [2,4,5].

There have been different monitoring programmes and research works in the coastal areas of Fiji; however, there have been no works that have attempted to study the magnitude of the influence from different variables, including climatic changes and anthropogenic inputs [6–10]. Modern natural resource sustainability plans seem to have shifted focus from concentrating on single-species management to a more inclusive and effective ecosystem-based management approach [11–13]. Information gathered from socioeconomic and scientific works needs to be integrated to form a baseline against which a long-term strategic monitoring programme can be set up at critical sites. Annual data collection over time will aid in resilience building, monitoring and tracking the effectiveness of implemented management plans.

Conventional field-based work often collects information about fractionated areas of large and complicated spatial cover with heterogeneous distribution. This makes it difficult to sufficiently encompass the spatial complexity of various biotic and abiotic components. Such works have poor capacity to monitor and detect small but important changes over time [14,15]. Coastal aquatic resources can be more effectively mapped using the Geographic Information System (GIS). It can be used to generate high-resolution spatial distribution maps of coastal resources or benthic habitats [6,16–19]. 'A benthic habitat is defined as an area of the seabed that is distinct from its surrounding in terms of physical, biological, and chemical variables' [13]. Habitat mapping can be defined as 'the use of spatially continuous environmental data sets to represent and predict biological patterns on the seafloor (in a continuous or discontinuous manner' [20]. Such maps have a high degree of reliability and serve as baselines. Frequent and regular generation of maps will allow monitoring and effective management of coastal resources. Marine protected areas (MPAs) have traditionally been set up in the exclusive economic zones (EEZs) of PICs using mostly ad hoc approaches. Such approaches are usually not based on any form of researched information [21]. Failure and poor effectiveness of numerous MPAs result from such set-ups. Maps that show high-resolution spatial distribution of coastal aquatic resources can be used as a baseline to set up MPAs, which can also be effectively monitored through regular generation of maps over time. Local knowledge integration with benthic resource maps allows for multifaceted information, which can be used by coastal mangers, as well informed baseline information for coastal resource management and MPA set-ups [6,22–25]. Habitat maps provide visual representation of the resources and how different components are linked. An inventory of the resources can be tracked, and threats can be identified [17–19,25]. Resource exploitation can also be planned to reduce pressure from sensitive areas.

For effective ecosystem-based management approaches, complex forecasting models are needed, which is possible through the GIS with in situ data collection and high-resolution satellite imaging [26–29]. Understanding the resource baseline, decision making and management planning approaches are hindered without such maps. As explained in Mumby et al. [30], conventional field-based approaches are less economical and time consuming in comparison to GIS tools [6]. PICs are normally constrained in terms of resource availability, and GIS methods will be of significant advantage. This work was intended to provide the village of Votua with a baseline benthic cover map of their resource-harvesting area and MPA area for effective management. This study was aimed at developing reliable benthic cover maps of village intertidal resource harvest areas (Cakaunisasi and Yarawa reefs) and anthropogenic perceptions of Votua Village in the Ba region of Fiji for better resource management planning and monitoring.

## 2. Materials and Methods

### 2.1. Study Site

The study was conducted on two reef patches named locally as the Cakaunisasi reef and the Yarawa reef, covering an area of ~63 km². These reefs are collectively called Votua Reef, which forms part of the Great Sea Reef in Fiji (locally known as Cakaulevu), which is the world's third-longest continuous barrier reef system [31]. Votua Reef is a shallow water ecosystem and becomes partially exposed at low tide. It is located in the north-eastern area, around 5km offshore from the main island of Viti Levu in Fiji (Figure 1). The study area also consisted of a marine protected area (MPA), as shown in Figure 1. This area is managed by the village of Votua located in the Nailaga district of the Ba region in Viti Levu. The study site was located between 177°31′09″ E and 177°40′60″ E longitude and 17°20′59″ S and 17°27′16″ S latitude.

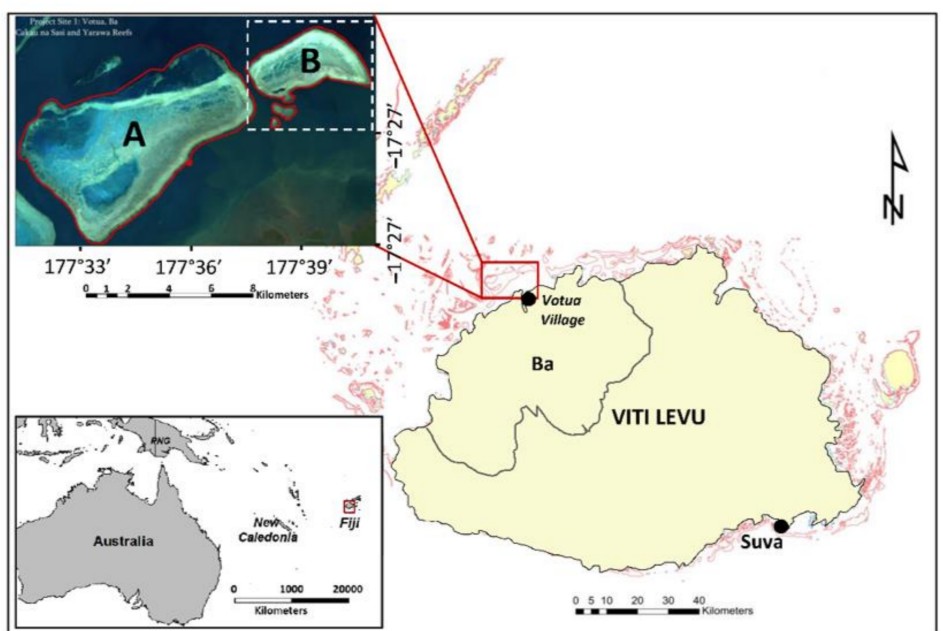

**Figure 1.** Map showing the study site location at Votua located in the Nailaga district of the Ba region, situated in the north-eastern coastal region of Viti Levu, which is the main island of Fiji. (**A**) Cakaunisasi reef (~50.8 km²) and (**B**) Yarawa reef (12.0 km²). Dashed line—marine protected area (MPA).

The coastal and inshore areas in Votua, including the Votua reef system, is traditionally owned and managed by Votua Village. The village comprises 150 households with an approximate population of 890 individuals, of which about 430 are males [32]. Drawing from the experiences of previous indigenous community-based studies, this study used interviews in an attempt to obtain more insight into the community perception of climate change and MPAs.

### 2.2. Imagery Used

A single WorldView2 satellite image of the study site was obtained and utilised. The image was dated on 24 January 2013, acquired from the Institute of Marine Remote Sensing located at the University of South Florida. The image had a ground resolution of 2 m. The image data were received in LV1B format, which is the most basic of the WorldView2 imagery product. The image was sensor- and radiometrically corrected; however, it was not projected using map projection [33].



### 2.3. Image Pre-Processing

Radiometric calibration was performed for the image using ENVI's (Environment for Visualizing Images) WorldView Radiance tool. Radiance values were produced through multiplication of the pixel value with the metadata-based gain and addition of the offset [34]. Radiometric correction of the image was performed using fast line-of-sight atmospheric analysis (FLASSH). FLAASH helps reveal information about at-the-surface reflectance properties of various benthic cover types that are present through reduction of atmospheric effects that are present on the imagery [35]. Ground reference points (GRPs) were aligned to the imagery by performing geometric corrections. Cropping of the study site was done from the entire imagery in order to maximise computational capacity. Areas that were spectrally insignificant were masked out of the image, for example, white-water areas that were on the reef crest.

### 2.4. Field Survey

A field survey was performed using an over-the-side drop video camera (SeaViewer 6000 HD Sea-Drop 1080 p camera) transect survey with slow trolling at high tide over a period of five days between October and November 2016. During the survey, a total of 60 videos were taken, ranging from 3 min to 15 min in length. The drop camera was equipped with a Global Positioning System (GPS) interface (Garmin GPSMAP 78) to record geographical coordinates in synchrony with the video frames, which were later differentially corrected. This was done by comparing the geographical coordinates with coordinates of known fixed positions. The GPS had an expected relative accuracy of 1–2 m. A field log sheet was also utilised for taking field notes.

### 2.5. Data Processing

Photos captured from the video transects were geotagged during the survey. This was done using the plugin EVIS Event Browser, which allows inputs of photos that are not geotagged and geographical coordinates from .csv files. This resulted in geotagged photos, which were imported into QGIS.

Photographs were categorised according to the habitat class present. In cases where habitat classes were mixed, photographs were classified according to the most dominant habitat type determined through qualitative assessment. A total of 250 GRPs were delineated from the video survey. These were randomly halved, with 125 GRPs used to train the algorithm for classification and the remaining GRPs used for accuracy assessment.

### 2.6. Supervised Classification and Habitat Map Production

QGIS is a free and open source software (managed by the Open Source Geospatial Foundation, OR, USA) [36] used for image processing. The semi-automatic classification plugin (SACP) [37] was used in QGIS for habitat map production. The plugin needs to be downloaded and installed from the Plugins repository in QGIS prior to the analysis. The SACP method assumes that there are distinct spectral signatures for different marine habitat types for multispectral satellite imagery. The light-reflective properties of different objects in a marine habitat are distinct from each other and dependent upon the biological and physical properties of the object. This reflectance at different wavelengths can be used to identify different habitats.

Training samples were created in the SACP using the geotagged images collected from the field exercise. The training samples were used for developing machine learning algorithms to define objects based on different spectral signatures. The pixels in training samples are categorised based on their maximum likelihood of belonging to a particular class of benthic cover (for this study) [38]. After machine learning, the SACP automatically categorises pixels into different classes based on previously defined criteria and algorithms. The SACP uses maximum likelihood [39], minimum distance [40] and spectral angle mapping [41] automatic algorithm types for the classification of an entire image. Images

generated from each algorithm were compared. The minimum distance algorithm was used to produce final habitat maps since it performed significantly better than other algorithms.

### 2.7. Accuracy Assessment

To gauge the uncertainty of the habitat maps created and carry out a quantitative assessment [42], an accuracy test was conducted for the classified habitat maps. A validation sample was created using 125 GRPs. Each classified pixel of the habitat map was compared against each classified pixel of the validation sample, and an accuracy assessment was obtained for each classified habitat map.

The kappa coefficient of the agreement value is frequently used as an image classification accuracy and reliability reference point. There are various useful features of the kappa coefficient, which makes it an attractive tool as an index of classification accuracy [43]. There are sometimes chance agreements in image classification and kappa coefficient compensates for this. This permits for calculation of a variance term which makes it possible for various statistical tests to be applied to test for significance between two coefficients [44]. Recent works have used the kappa coefficient as a tool for image accuracy and reliability [6,45–50].

### 2.8. Collection of Ancillary Data on Fishing Grounds

Publication and sharing of information from the research were part of an agreement between the University of Fiji, the Ministry of iTaukei (local Fijian term for individuals of native Fijian origin) Affairs (MIA), villagers and the research grant donors (United States Agency for International Development (USAID)). Representatives from the village were part of the data collection activities, and the villagers were updated regularly on the progress of the project.

Quota sampling was also used to identify 50 participants for in-depth semi-structured interviews. The categories of interest included gender, age and livelihood. A total of 28 males and 22 females in the age range of 20 to 70 years participated in the interviews. Approximately 80% of the interviewees engaged in fishing and invertebrate gleaning for livelihood. The rest were involved in some form of paid employment. The interview questionnaire included some questions that were designed to elicit responses of yes, no and no change, while a few open-ended questions focused on climate change perceptions, resource harvest over time, significance and status of MPA and benthic distribution of resources.

Additionally, a talanoa (an informal gathering where the researcher and respondents shared stories and research findings over a few bowls of kava (a popular social drink in the South Pacific Island countries made from the extract of the plant *Piper methysticum*)) session was organised towards the end of the research, and the village leaders and participants were invited to discuss the research findings. The villagers were asked to indicate the fishing sites, benthic cover distribution and spawning phenology of popular fish species on a printed map that was provided to them. This activity helped in the verification and fine-tuning of the maps generated. The information gathered via the verification process were added as GIS layers and made available to the villagers for sustainable management of the marine resources, which is the main source of their livelihood.

## 3. Results

Six main habitat classes were found, based on their dominance on the Votua reef system (Figure 2). Quantitative analysis was based on field observation and pictures collected with a drop camera. For pictures containing mixed habitat classes, the most dominant class in each picture was selected as the overall class.

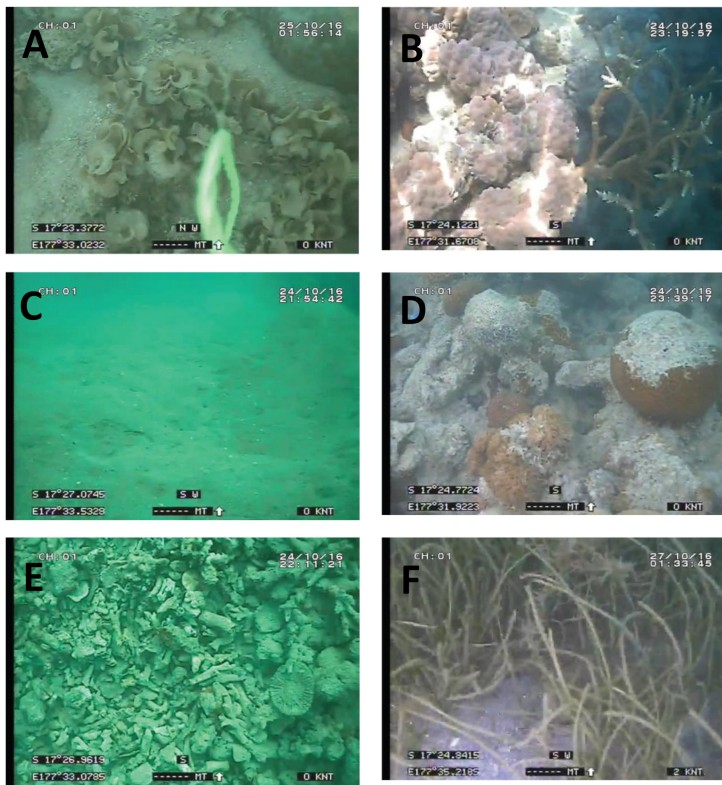

**Figure 2.** Six classes of benthic class types identified: (**A**) algae, (**B**) coral, (**C**) sand and gravel, (**D**) buried reef, (**E**) coral rubble and (**F**) seagrass.

Benthic habitats for the six identified classes are represented in Figure 3. The biotic cover consisted of mainly algae, coral, buried reef and seagrass. The algae cover comprised a mixture of green, red and brown algae. The coral class was represented only by living corals. The buried reef consisted of live corals that had been partially covered with a sediment layer. Seagrass was the most dominant biotic cover, followed by coral, buried reef and algae. The abiotic cover was represented by sand and gravel and coral rubble. Sand and gravel were a mixture of both submerged and exposed volcanic rocks. Coral rubble consisted of degraded and broken coral pieces. Sand and gravel had a higher coverage on the Yarawa reef compared to the Cakaunisasi reef of the Votua reef system (Figure 3A).

Results of the accuracy assessment for the six identified benthic cover types are summarised in Table 1 The full error matrix is presented in Table 2. From Table 1, the overall kappa hat classification value was 0.8, with an overall accuracy of 82.2%. The kappa value is used as an image classification reliability reference point. The kappa hat value arises as a result of conformity between the reference data and the image classification map [51]. The kappa value ranges from 0 to 1, with 1 showing perfect agreement and 0 showing no agreement. For other values, a kappa value of <0.20 is considered as poor, <0.40 as fair, <0.60 as moderate, <0.80 as good and >0.8 as very good agreement [52,53]. The kappa value was greater than 0.88 in all cases except for the class of coral rubble (Table 1). Coral rubble was incorrectly classified as seagrass.

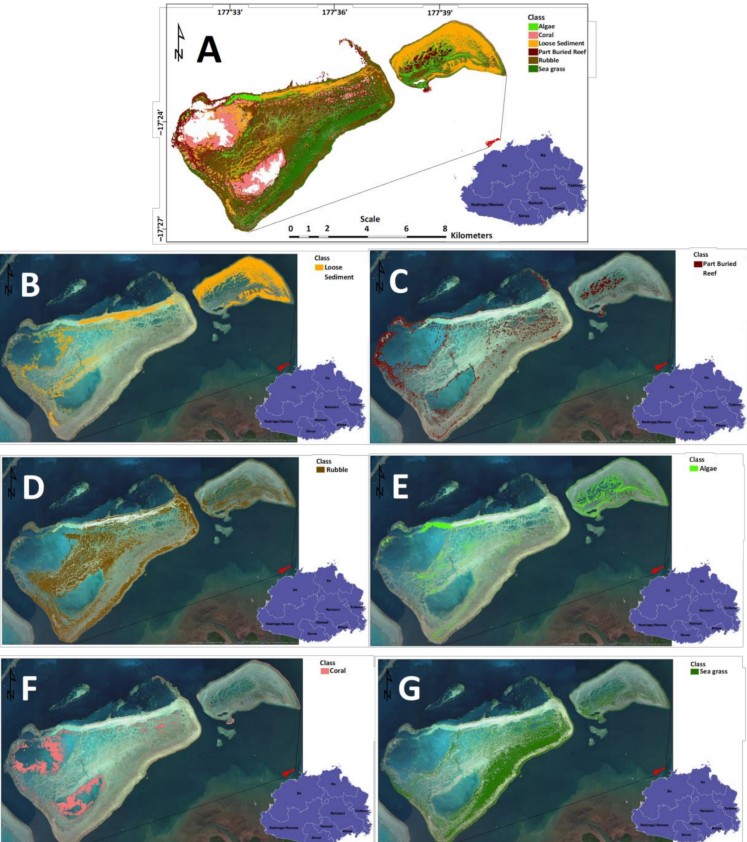

**Figure 3.** Six classes of habitat cover for the Cakaunisasi and Yarawa reefs of the Votua reef system, Ba, Fiji. Different letters show different attributes of the reef system benthic cover. (**A**) Overall benthic cover map showing six classes of habitat types and location, (**B**) sand and gravel, (**C**) buried reef, (**D**) coral rubble, (**E**) algae, (**F**) coral and (**G**) seagrass.

Table 3 shows the results of the socioeconomic survey from Votua Village on climate change perceptions and knowledge of the MPA zone in the study area. Based on questionnaire and *talanoa* sessions, the Cakaunisasi and Yarawa reef systems under study included important feeding zones, spawning areas and spawning periods for various fish species, which are of ecological, subsistence and commercial importance. These species include *Scomberomorus commerson* (Spanish mackerel), *Sphyraena barracuda* (great barracuda), *Cheilinus trilobatus* (tripletail wrasse), *Mugil cephalus* (grey mullet), *Macrobrachium lar* (monkey river prawn), *Rastrelliger brachysoma* (short mackerel), *Trochus niloticus* (trochus), *Trochus pyramis* (top shell), *Spondylus ducalis* (thorny oyster) *Panulirus penicillatus* (spiny lobster), *Katsuwonus pelamis* (skipjack tuna), *Thunnus albacares* (yellowfin tuna), *Thunnus alalunga* (albacore tuna), *Thunnus obesus* (bigeye tuna), *Holothuria scabra* (sandfish), *Pristipomoides* spp. (snapper/jobfish) and *Batissa violacea* (freshwater clam). Information about these species was included on maps shared with Votua Village for use in its management plans.

**Table 1.** The accuracy assessment and results of the confusion matrix for the six benthic cover class types, including the overall accuracy and kappa coefficient for the benthic cover maps of the two reefs of the Votua reef system in the Ba region of Fiji.

| Class | Producers' Accuracy (%) | Users'Accuracy (%) | Kappa Hat Classification | Overall Kappa Hat Classification | Overall Accuracy (%) | Percentage Cover Cakaunisasi (%) | Percentage Cover Yarawa (%) | Percentage Total Cover (%) |
|---|---|---|---|---|---|---|---|---|
| Sand and gravel | 75.82 | 100.00 | 1.00 | 0.80 | 82.2 | 12.69 | 46.41 | 19.90 |
| Coral | 95.16 | 100.00 | 1.00 | | | 11.82 | 2.93 | 10.00 |
| Coral rubble | 82.76 | 45.28 | 0.39 | | | 33.20 | 19.76 | 30.30 |
| Algae | 100.00 | 93.83 | 0.93 | | | 4.41 | 14.64 | 6.60 |
| Seagrass | 64.03 | 90.82 | 0.88 | | | 24.43 | 8.61 | 21.00 |
| Buried reef | 98.77 | 90.91 | 0.89 | | | 13.45 | 7.65 | 12.20 |

**Table 2.** The confusion error matrix for the six benthic cover class types.

| Class | Merged Classes | Sand and Gravel | Coral | Coral Rubble | Algae | Seagrass | Buried Reef | Total |
|---|---|---|---|---|---|---|---|---|
| Merged classes | 0 | 18 | 3 | 0 | 0 | 0 | 0 | 21 |
| Sand and gravel | 0 | 116 | 1 | 22 | 15 | 18 | 2 | 116 |
| Coral | 0 | 0 | 75 | 0 | 0 | 0 | 0 | 59 |
| Coral rubble | 0 | 0 | 0 | 0 | 1 | 0 | 0 | 106 |
| Algae | 0 | 0 | 59 | 0 | 0 | 0 | 0 | 81 |
| Seagrass | 0 | 8 | 0 | 48 | 0 | 50 | 0 | 98 |
| Buried reef | 0 | 5 | 0 | 0 | 76 | 0 | 0 | 88 |
| Total | 0 | 153 | 62 | 58 | 76 | 139 | 81 | 569 |

**Table 3.** Interviewees' perceptions of climate change and Marine Protected Area.

| Question | Yes (%) | No (%) | No Change (%) | Total Respondents |
|---|---|---|---|---|
| Has the frequency of rainfall significantly increased over the past ten years? | 94 | 2 | 4 | 50 |
| Has the frequency of coastal flooding significantly increased over the past ten years? | 88 | 4 | 8 | 50 |
| Has the frequency of violent storms significantly increased over the past ten years? | 94 | 0 | 6 | 50 |
| Has the frequency of drought events significantly increased over the past ten years? | 86 | 8 | 6 | 50 |
| Over the past ten years, has the summer season temperature: | | | | |
| Increased? | 88 | 4 | 8 | 50 |
| Decreased? | 4 | 90 | 6 | 50 |
| Over the past ten years, has the winter season temperature: | | | | |
| Increased? | 56 | 32 | 12 | 50 |
| Decreased? | 40 | 48 | 12 | 50 |
| Has the wave energy increased significantly over the past ten years? | 80 | 6 | 14 | 50 |
| Has the amount of your resource harvest per harvest event declined over the past ten years? | 96 | 0 | 4 | 50 |
| Do you know the reason for the set-up of the MPA? | 78 | 22 | - | 50 |
| Was the MPA set-up to restore the harvestable resources in the particular area? | 100 | 0 | - | 50 |
| Do you know what baseline information was used for the MPA set-up? | 12 | 88 | - | 50 |
| Were there scientific consultations done for the MPA set-up? | 14 | 86 | - | 50 |
| Is the MPA performing as well as expected? | 16 | 84 | - | 50 |
| If no, is the poor MPA performance due to: | | | | |
| Climate change effects? | 86 | 14 | - | 50 |
| Anthropogenic activities? | 22 | 78 | - | 50 |
| Do you think a baseline coastal and land use resource map will help in making better resource management plans? | 94 | 0 | 6 | 50 |

All maps generated in this study including some not shown here are hosted as online interactive maps.

## 4. Discussion

Reliant baseline information or habitat maps provide important information about the status and spatial distribution of natural resources. This is important for having reference points and gauging the effectiveness of implemented management plans. Regular mapping of the same area over time helps to monitor the status of the resources and demonstrates trends and patterns arising over time. Such information helps managers formulate targeted management plans in affected areas and, consequently, monitor the effectiveness of the management plans. Effects of tourism activities, natural disasters, anthropogenic influences and pollution from point and non-point sources, among other variables, can be monitored and managed over time using GIS spatial mapping techniques [54–60].

This study aimed at developing reliable benthic cover maps of village intertidal resource harvest areas and anthropogenic perceptions of Votua Village for better resource management planning and monitoring. WorldView2 imagery used here has a high level of accuracy; however, it is quite expensive. Roelfsema and Phinn [61] compared the economic viability of eight different field and remote sensing approaches for mapping the benthic community cover for different coastal and reef systems in Fiji. Using Landsat imagery was determined as one of the more economical approaches in terms of time and cost; however, it offers a moderate level of accuracy. The hydroacoustic method is also an economical option for high-resolution aquatic habitat mapping with reliable accuracy; however, it is recommended for use in integration with other methods [62,63]. A combination of hydroacoustic and Landsat methodologies can be used as a more economical option for regular map generation in long-term monitoring works, and WorldView2 can be considered as an investment for accurate baseline mapping.

The Yarawa reef, which is part of the Votua Village MPA, is completely managed by the villagers. MPAs set up are reserves of aquatic flora and fauna resources that have been defined by law or other effective means to protect or allow recovery of resources within the enclosure and, with suitable monitoring plans, have good potential for adaptation and mitigation against climate change influences [64–66]. The baseline map developed in this study is a reference point for the status and benthic resource distribution for the MPA. Regular mapping is needed to monitor the effectiveness of the MPA management approach over time. Small changes can also be monitored and management plans improved if the MPA approach does not perform as anticipated. According to the participants, the neighbouring Cakaunisasi reef is not an MPA, and regular resource exploitation takes place, including fishing activities. From the maps generated in Figure 3 and Table 1, the Cakaunisasi reef (non-MPA) has a better coral and seagrass cover compared to the Yarawa reef (MPA). Most participants indicated that MPA set-up did not have any prior scientific consultations. Biological information forms one of the important criteria for MPA set-up and provides a better chance of success [65]. Regular generation of maps of the two reefs will enable a better understanding of the processes involved, leading to better decisions. Benthic cover maps are important in setting up MPAs. A study conducted in the Caribbean demonstrated that estimation of different benthic species is useful for the set-up and monitoring of MPAs [67].

The participants showed some level of understanding of the impacts of climate change. This was due to the participants' own observation of the local indicators of environmental change and the increasing number of awareness programmes conducted by the government and non-governmental organizations. A majority of the respondents stated that climate change adversely affected the performance of the existing MPA and also contributed to a decline in the abundance of harvestable marine species such as *C. trilobatus*, *M. cephalus* and *Pristipomoides* spp. A small number of respondents related the poor state of the MPA to anthropogenic-related activities. It is evident that the villagers' set up an MPA to restore harvestable resources; however, they lacked the baseline information that was

required for the set-up. The limitation of scientific consultation may have been one of the factors resulting in the poor performance of the present MPA. There were various areas of ecological importance identified by the villagers, which included feeding and spawning sites for various harvestable fish species. This information, along with fishing sites and benthic cover maps, will form important baseline data for the villagers as well as the MIA to consider an appropriate revised zone for the MPA.

This study revealed that there is a lack of reliable data on historical coastal resource harvesting for the two reefs in Votua. This creates data gaps that prevent scientists from carrying out resource assessments, which is not possible without some form of reliable time-series data. To partially fulfil this requirement, information can be gathered from various fishermen through suitably designed questionnaires and socioeconomic surveys. The data gathered may not be as reliable as scientific data; however, it can help gauge the past and present situations of the resources. Such information can be integrated with resource mapping for formulation of better management plans.

The baseline maps generated through this study have sufficient accuracy to be of reliable quality (Table 1). These maps can be integrated with monitoring of other parameters such as coastal land use activities, water quality assessment, anthropogenic actions and activities along the watershed leading to the waters surrounding the two reefs. These data can be integrated in baseline maps to form ecosystem maps, which will show different variable layers and factors affecting a marine ecosystem on a single platform to aid in better informed decision making such as MAP formation. Such an approach can form the basis of ecosystem-based management techniques and aid in identifying the magnitude of the influences of human activities and changing climatic conditions. This is especially important since human-related influences can be regulated with greater ease in comparison to managing climatic variables.

## 5. Conclusions

The maps generated in this study provide baseline information about resource distribution on Cakaunisasi and Yarawa reefs. Enrichment of the maps with other parameters such as water quality, coastal land use activities, fishery activities and socioeconomic information will allow for the formation of ecosystem maps. The existing MPA on the Yarawa reef does not seem to be working as well as anticipated. Baseline maps constructed here and possibly ecosystem maps can allow for monitoring the existing MPA as well as forming a new and more informed MPA. A sustained monitoring programme with regular map generation is needed to better formulate and monitor management approaches, inform policy decisions and develop community resilience against projected changes towards the abundance and distribution of resources.

**Author Contributions:** Conceptualization, A.A.S. and A.M.; formal analysis, A.A.S. and A.M.; funding acquisition, P.S.; investigation, A.A.S., A.M. and P.S.; methodology, A.A.S. and A.M.; project administration, A.M.; resources, A.A.S. and P.S.; software, A.M.; supervision, A.M.; writing—original draft, A.A.S.; writing—review and editing, A.A.S. and P.S. All authors have read and agreed to the published version of the manuscript. This work was carried out in collaboration between all authors.

**Funding:** This research was funded by the United States Agency for International Development (USAID), grant number PACAM-14-0002.

**Institutional Review Board Statement:** Not applicable.

**Informed Consent Statement:** Informed consent was obtained from all subjects involved in the study.

**Data Availability Statement:** All the maps generated using the GIS and satellite imagery in this study for Cakaunisasi and Yarawa reefs of the Votua reef system in Votua, Nailaga District, Ba, Fiji, are hosted as interactive maps and maintained by the University of Fiji. The maps contain benthic cover and local knowledge layers. These maps serve as baseline references for the measurement of changes in the future for sustainability and management purposes. The following link contains a guide for how to use the web map application. The .zip file needs to be downloaded and unzipped

to access the guide: https://www.unifiji.ac.fj/pacam/#1524118889762-66e09b4d-a92f. The following link takes you to the interactive maps for the Votua project site: https://pacamuof.maps.arcgis.com/apps/webappviewer/index.html?id=09ff6d87bc0c455093449779853875b1. The following link takes you to the static maps for the Votua project site: https://pacamuof.maps.arcgis.com/sharing/rest/content/items/bfe701012c0f482e9ef9e8e38659bdb4/data.

**Acknowledgments:** This work was made possible through the funding provided by the United States Agency for International Development (USAID) to the University of Fiji through the Pacific–American Climate Fund (PACAM) for the project Developing Base Maps of Tropical Aquatic Resources in the Pacific. The authors acknowledge the following for the support provided: USAID, PACAM, the University of South Florida, the University of Fiji, Salesh Kumar and Zulfikar Begg of the Geoscience Division at Pacific Community (SPC) in Fiji, the Ministry of iTaukei Affairs (Fiji), Ministry of Education (Fiji), Fisheries Department (Fiji), Michelle Kumar and the residents of Votua Village for their assistance, cooperation and hospitality.

**Conflicts of Interest:** The authors declare no conflict of interest. The funders had no role in the design of the study; in the collection, analyses or interpretation of data; in the writing of the manuscript; or in the decision to publish the results.

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
