# Peer review of "Benthic Resource Baseline Mapping of Cakaunisasi and Yarawa Reef Ecosystem in the Ba Region of Fiji"

_water, doi:10.3390/w13040468_

Round 1
Reviewer 1 Report
Without any doubts, benthic habitat mapping in such valuable areas as Fiji coastal area is of significant content for establishment and monitoring of Marine Protected Areas. Authors used simple remote sensing methods for supervised classification based on satellite images and ground-truth dataset. However, some following remarks should be explained before the acceptance of the manuscript.
Consider careful proofreading of your manuscript and adjusting the text to more scientific journal style - strict and brief, without unnecessary details.
In the Introduction, it would be appropriate if you will state the proper definition of benthic habitat mapping. See Brown et al., 2011; Lecours et al., 2015
Brown, C.J.; Smith, S.J.; Lawton, P.; Anderson, J.T. Benthic habitat mapping: A review of progress towards improved understanding of the spatial ecology of the seafloor using acoustic techniques. Estuarine, Coastal and Shelf Science 2011, 92, 502-520, doi:10.1016/j.ecss.2011.02.007.
Lecours, V.; Devillers, R.; Schneider, D.C.; Lucieer, V.L.; Brown, C.J.; Edinger, E.N. Spatial scale and geographic context in benthic habitat mapping: review and future directions. Marine Ecology Progress Series 2015, 535, 259-284, doi:10.3354/meps11378.
Figure 1. Strange way of representing the inset situation map. Could you remove the continuous radial lines extending from the upper left corner of the subset map? They are unnecessary covering countries boundaries at the map. The scale is needed for this map as well.
lines 117-126: Please, provide details of satellite dataset, like i.e. spatial resolution
lines 128-133: Provide accuracy of GPS measurements, underwater camera positioning method, and model of a used camera.
line 139: did you applied any habitat classification scheme? See Strong et al., 2019
Strong, J.A.; Clements, A.; Lillis, H.; Galparsoro, I.; Bildstein, T.; Pesch, R.; Birchenough, S. A review of the influence of marine habitat classification schemes on mapping studies: inherent assumptions, influence on end products, and suggestions for future developments. ICES Journal of Marine Science 2019, 76, 10-22, doi:10.1093/icesjms/fsy161.
lines 156-157: provide literature references for each of used method of supervised classification
line 158: what was the source of the reference shape that you mentioned?
line 165: what was the source of validation subset?
line 247-248 - Consider Lucieer et al., 2013 for further explanations of interpretation of Kappa accuracy statistics.
Lucieer, V.; Hill, N.A.; Barrett, N.S.; Nichol, S. Do marine substrates ‘look’ and ‘sound’ the same? Supervised classification of multibeam acoustic data using autonomous underwater vehicle images. Estuarine, Coastal and Shelf Science 2013, 117, 94-106, doi:10.1016/j.ecss.2012.11.001.
line 249-250: provide a proper reference to Kappa and its explanation in methodology. Consider Foody, 2002. Be careful, because some authors question the validity of using Kappa in the literature (see. Pontius et al., 2011)
Foody, G.M. Status of land cover classification accuracy assessment. Remote Sensing of Environment 2002, 80, 185-201, doi:10.1016/s0034-4257(01)00295-4.
Pontius, R.G.; Millones, M. Death to Kappa: birth of quantity disagreement and allocation disagreement for accuracy assessment. International Journal of Remote Sensing 2011, 32, 4407-4429, doi:10.1080/01431161.2011.552923.
Table 1. To be more transparent, provide the complete error matrix.
lines 277-284: This is true. In this part, it is worth citing the recent paper about benthic habitat change detection in the critical site of Venice inlet from Janowski et al., 2020:
Janowski, L.; Madricardo, F.; Fogarin, S.; Kruss, A.; Molinaroli, E.; Kubowicz-Grajewska, A.; Tegowski, J. Spatial and Temporal Changes of Tidal Inlet Using Object-Based Image Analysis of Multibeam Echosounder Measurements: A Case from the Lagoon of Venice, Italy. Remote Sensing 2020, 12, 2117, doi:10.3390/rs12132117.
lines 288-294: The mention of costs is often neglected by other authors. It would be appropriate if you could compare the costs of satellite images with hydroacoustic surveys, which are becoming the general method for high-resolution benthic habitat mapping.
Author Response
An amended document has been prepared with track changes shown. Responses are in reference to the amended document.
Point 1: In the Introduction, it would be appropriate if you will state the proper definition of benthic habitat mapping. See Brown et al., 2011; Lecours et al., 2015
Response 1: Proper definition of benthic habitat mapping was included in the introduction in lines 68-72.
Point 2: Figure 1. Strange way of representing the inset situation map. Could you remove the continuous radial lines extending from the upper left corner of the subset map? They are unnecessary covering countries boundaries at the map. The scale is needed for this map as well.
Response 2: Figure 1. The continuous radial lines extending from the upper left corner of the subset map were removed and a scale was added for this map as well. Amended map is included in line 112-113.
Point 3: lines 117-126: Please, provide details of satellite dataset, like i.e. spatial resolution
Response 3: spatial resolution of image was added in lines 127-128.
Point 4: lines 128-133: Provide accuracy of GPS measurements, underwater camera positioning method, and model of a used camera.
Response 4: Accuracy of GPS measurements, camera positioning method and model of both GPS and camera was included in lines 144-150.
Point 5: line 139: did you applied any habitat classification scheme? See Strong et al., 2019
Response 5: Habitat classification scheme was included in lines 159-162.
Point 6: lines 156-157: provide literature references for each of used method of supervised classification
Response 6: Literature reference were added for each classification method in lines 180-181.
Point 7: line 158: what was the source of the reference shape that you mentioned?
Response 7: This sentence was deleted to avoid confusion. This was redundant. Image validation and accuracy assessments is already included in section 2.7, lines 186-190.
Point 8: line 165: what was the source of validation subset?
Response 8: Validation subset source was added in lines 187-188.
Point 9: line 247-248 - Consider Lucieer et al., 2013 for further explanations of interpretation of Kappa accuracy statistics.
Response 9: Kappa statistics was explained further in lines 192-198 with references included.
Point 10: line 249-250: provide a proper reference to Kappa and its explanation in methodology. Consider Foody, 2002. Be careful, because some authors question the validity of using Kappa in the literature (see. Pontius et al., 2011)
Response 10: Kappa statistics was explained further in lines 192-198. Latest works using kappa statistics for accuracy and reliability were included to justify its use.
Point 11: Table 1. To be more transparent, provide the complete error matrix.
Response 11: Error matrix was added in line 307 as Table 1B.
Point 12: lines 277-284: This is true. In this part, it is worth citing the recent paper about benthic habitat change detection in the critical site of Venice inlet from Janowski et al., 2020:
Response 12: In addition to Janowski et al., 2020, other latest references were also cited, shown in line 335.
Point 13: lines 288-294: The mention of costs is often neglected by other authors. It would be appropriate if you could compare the costs of satellite images with hydroacoustic surveys, which are becoming the general method for high-resolution benthic habitat mapping.
Response 13: Economics of satellite images with hydroacoustic surveys was incorporated with references in lines 343-346.
Reviewer 2 Report
Translator
Translator
Suggestions and comments are included in the attached .pdf as notes.
The paper is strong in the satellite benthic interpretation.However, I recomend to change the legend of the figure. The habitat names should follow the habitat classification used in the literature. The terms 'Loose sediment' or 'rubble' shoud be changed, 'sea grass' should be changed into 'seagrass'.
The part regarding the people interviews is interesting but it lacks of one scientific treatment of data. Conclusions about the management of MPA and not-MPA are only general statements. You have a detailed cartography, and interviews with presence/absence of species, but you cannot publish them, nor put them on the maps. Hence, these informations may be of interest in one paper corcening the fishing and/or biodiversity of Fiji or one on social and economic use of fishing or of MPAs, but it not appear well integrated in the structure of the paper.

Author Response
Point 1: Suggestions and comments are included in the attached .pdf as notes.
Response 1: All track changes made by reviewer were accepted and incorporated in the document. Responses to all other comments, suggestions and queries from the pdf file are shown in point form, from point 3 below.
Point 2: The paper is strong in the satellite benthic interpretation. However, I recomend to change the legend of the figure. The habitat names should follow the habitat classification used in the literature. The terms 'Loose sediment' or 'rubble' shoud be changed, 'sea grass' should be changed into 'seagrass'.
Response 2: For Figure 2 and Figure 3 legends were changed. The term 'Loose sediment' was changed to 'sand and gravel', 'part buried reef' to 'buried reef', 'rubble' to 'coral rubble' and 'sea grass' was changed into 'seagrass'.
Point 3: please correct the black and white figure. ( delete the lines /rays from the top right vertex ) Line 105, Figure 1. Also the red square the two red lines in the figure shoud be deleted.
Response 3: The black and white figure was corrected and the red lines were deleted shown in Figure 1, line 120.
Point 4: What do you mean for 'differentially corrected'? line 132. please explain
Response 4: A sentence was added to explain this in lines 149-151.
Point 5: I did not find the SACP plugin in the QGIS menu. Perhaps it is available only for one QGIS version? It is not clear if QGIS is free or it is the plugin. Please clear if the plugin is available. lines 144-145.
Response 5: Availability of the SACP plugin was explained in lines 166-169.
Point 6: how many GRPs were used ? line 164.
Response 6: Number of GRPs were added in line 189.
Point 7: Add some references line 285.
Response 7: References were added in line 336.
Point 8: Please change the sentence. what do you mean with 'larger and smaller' reefs .?Perhaps Cakaunisasi reef and Yarawa reef ? Lines 238-239.
Response 8: This is correct. The reef names were added in line 282.
Point 9: Line 307-308. To write this conclusion you should at least add a table where habitat percentage covers are compared among the two islands
'better biotic cover' doesn't mean anything.
Response 9: Line 307. Habitat percentage cover for each site was incorporated in Table 1A.
Round 2
Reviewer 1 Report
It is good to see that the vast majority of concerns were addressed properly. Therefore, I recommend the paper for acceptance.